# Side Chain-Modified Benzothiazinone Derivatives with Anti-Mycobacterial Activity

**DOI:** 10.3390/biomedicines11071975

**Published:** 2023-07-12

**Authors:** Dongguang Fan, Bin Wang, Giovanni Stelitano, Karin Savková, Olga Riabova, Rui Shi, Xiaomei Wu, Laurent R. Chiarelli, Katarína Mikušová, Vadim Makarov, Yu Lu, Yuzhi Hong, Chunhua Qiao

**Affiliations:** 1College of Pharmaceutical Sciences, Soochow University, Suzhou 215123, China; 2Beijing Key Laboratory of Drug Resistance Tuberculosis Research, Department of Pharmacology, Beijing Tuberculosis and Thoracic Tumor Research, Beijing Chest Hospital, Beijing 101149, China; 3Department of Biology and Biotechnology, University of Pavia, 27100 Pavia, Italy; 4Department of Biochemistry, Faculty of Natural Sciences, Comenius University in Bratislava, 84215 Bratislava, Slovakia; 5Research Center of Biotechnology of the Russian Academy of Sciences, Moscow 119071, Russia; 6Institute of Molecular Enzymology, School of Biology and Basic Medical Sciences, Suzhou Medical College, Soochow University, Suzhou 215123, China; 7Suzhou Key Laboratory of Pathogen Bioscience and Anti-Infective Medicine, Suzhou Medical College, Soochow University, Suzhou 215123, China

**Keywords:** anti-tubercular agents, DprE1 inhibitor, structure activity relationship, in vivo activity

## Abstract

Tuberculosis (TB) is a leading infectious disease with serious antibiotic resistance. The benzothiazinone (BTZ) scaffold PBTZ169 kills *Mycobacterium tuberculosis* (Mtb) through the inhibition of the essential cell wall enzyme decaprenylphosphoryl-β-D-ribose 2’-oxidase (DprE1). PBTZ169 shows anti-TB potential in animal models and pilot clinical tests. Although highly potent, the BTZ type DprE1 inhibitors in general show extremely low aqueous solubility, which adversely affects the drug-like properties. To improve the compounds physicochemical properties, we generated a series of BTZ analogues. Several optimized compounds had MIC values against Mtb lower than 0.01 µM. The representative compound **37** displays improved solubility and bioavailability compared to the lead compound. Additionally, compound **37** shows Mtb-killing ability in an acute infection mouse model.

## 1. Introduction

The notorious tuberculosis (TB) is listed as one of the deadliest infectious diseases worldwide. The reported mortality is 1.4 million people each year, and the estimated latent infection is 2 billion people [1]. Treatment of the disease using the first-line anti-tubercular agents, such as isoniazid, rifampicin, ethambutol and pyrazinamide, is challenged as more and more reported cases of multidrug resistant (MDR) and extensively drug-resistant (XDR) clinical mutants emerge [2]. To combat with the disaster of MDR- and XDR-TB, conscious and concerted efforts have continued to discover anti-TB antibiotics with new targets or scaffolds over the past decades.

A class of compounds with a benzothiazinone (BTZ) scaffold, first reported in 2009, show potent antitubercular activity and inhibit of the essential cell wall biosynthesis enzyme decaprenylphosphoryl-*β*-D-ribose 2′ oxidase (DprE1) [3,4,5]. Among all reported BTZ derivatives, BTZ043 and PBTZ169 [6] (Figure 1) attracted most attention because of their remarkable whole-cell activity, with the minimal inhibitory concentration (MIC) values lower than 0.004 µM. On top of this, the BTZ type candidates demonstrated synergistic effect with other TB drugs [6]. Reported studies have shown that the BTZ type DprE1 inhibitors are active against clinical MDR and XDR strains [3]. Although both benzothiazinones have been advanced to clinical trials, each suffered suboptimal drug-like properties. In addition to their low aqueous solubility, the presence of a chiral center in BTZ043 renders its synthesis disadvantageous; the more potent second generation PBTZ169 takes liability of poor in vivo bioavailability, presumably emanating from its low water solubility and high plasma protein binding fraction [7].

To improve the BTZ type compounds physicochemical properties, we herein report our investigation of novel series of derivatives with improved aqueous solibility and good pharmacokinetic activity. We demonstrated side chain modification by opening the cyclic piperidinyl or piperazinyl ring could expand the BTZ compound diversity, afford BTZ analogs with elevated physicochemical properties.

## 2. Materials and Methods

### 2.1. General Experimental Information

Reagents, solvents and materials were purchased from commercial suppliers and were used directly, unless further treatment was noted. Anhydrous solvent tetrahydrofuran (THF) and dichloromethane (DCM) were also obtained and used as commercial sources. Thin layer chromatography (TLC) was monitored and performed on silica HSGF254 plates to follow the reactions. TLC was virtualized under ultraviolet light (UV) light at 254 or 365 nm, or exposure to I_2_. The crude product purification was conducted on column chromatography silica gel (300–400 mesh). All final products were recorded via ^1^H NMR,^13^C NMR on an Agilent–400 MHz or Bruker DD2-600 MHz spectrometer in CDCl_3_ or dimethyl sulfoxide (DMSO-*d*_6_), TMS was used as reference. High resolution mass analysis (HRMS) in electrospray ionization mode was determined on Micromass GCT-TOF. The sample purity analysis was determined via high performance liquid chromatography (HPLC) on SHIMADZU LC–20AD system. Before sending for biological test, purity of all tested compounds was greater than 95%. 

### 2.2. Chemistry

#### Chemical Synthesis

Compounds **1**–**30** were prepared as shown in Figure 1. The intermediate **43** and **45** was prepared according to the published methods [8]. Click chemistry between **45** and corresponding azide using CuSO_4_ and sodium ascorbate in EtOH/H_2_O yield the final compounds **1**–**30**.

Reagents and conditions: (a) oxalyl chloride, a drop of DMF, DCM, 0.5 h, then, NH_4_SCN, DCM/CH_3_COCH_3_ = 5:1, 1.0 h, rt; (b) *N*-methylpropagyl amine, 2.5 h, DCM, yield: 65%; (c) corresponding azide R-N_3_, CuSO_4_, K_2_CO_3_, sodium ascorbate, EtOH/H_2_O, rt, 16 h, yield: 40–68%.

Compounds **31**–**36** were prepared as shown in Figure 2. The reaction intermediates **46a**–**c** and **47a**–**c** were prepared according to the published procedure [8]. The final cyclization reaction between **44** and the above intermediates yielded compounds **31**–**36** in medium yield.

Compounds **37**–**43** were prepared as shown in Figure 3. The reaction intermediates **48a**–**c** and **49a**–**c** were prepared according to the published procedure [8]. The final cyclization reaction of **44** with the above intermediates provided the target compounds **37**–**42** in 30–50% yields.

### 2.3. MIC Determination

The MIC against replicating M. tuberculosis against H37Rv were determined using microplate alamar blue assay (MABA), as previously reported [9]. Compound PBTZ169 was employed as a positive control. At 37 °C, *M. tuberculosis* H37Rv (ACTT 25618) strains were cultured in Difco Middlebrook 7H9 Broth (Seebio), with 0.2% (*v*/*v*) glycerol, 0.05% Tween 80, and 10% (*v*/*v*) albumin-dextrose-catalase (Seebio) (7H9-ADC-TG) to grow into late log phase (70–100 Klett units), then centrifuged, washed twice, and resuspended in phosphate-buffered saline. All tested compound stock solutions were initially prepared in DMSO, then serially diluted in two-fold in 7H9-ADC-TG in a volume of 100 μL in 96-well clear-bottom microplates (BD), then the bacterial culture suspension in 100 μL containing 2 × 10^5^ CFU) was added to yield a final testing volume of 200 μL. The tested compound concentration range was from 2 to 0.002 μg/mL. The plates containing the corresponding compounds were incubated at 37 °C. On day 7, 20 μL of alamar blue and 50 μL 5% Tween 80 were added to all wells. After incubation at 37 °C for 16–24 h, the fluorescence signal was read at an excitation of 530 nm and an emission of 590 nm. The MIC was defined as the lowest concentration resulting in a reduction in fluorescence of ≥90% relative to the mean of replicate bacterium-only controls.

### 2.4. Compound IC_50_ Determination

The recombinant *M. tuberculosis* DprE1 enzyme was produced in *E. coli*, as previously reported [10]. Enzymatic activity was assayed using a Amplex Red/peroxidase coupled assay at 30 °C. Briefly, DprE1 (0.15 μM) was incubated in 20 mM glycylglycine pH 8.5, containing 0.05 mM Amplex Red, and 0.35 μM horseradish peroxidase; the reaction was started by addition of 0.5 mM FPR, and monitored by measuring the generation of resorufin at 572 nm (ε = 54,000 M^−1^ cm^−1^). For inhibition studies, each compound (dissolved in DMSO) was firstly assayed at a final concentration of 20 μM, using DMSO as negative and PBTZ169 as positive control. Compounds displaying less than 20% of residual activity at 20 μM were further investigated through the determination of IC_50_ and the analysis of kinetic inactivation of DprE1, as previously reported.
(1)$$A_{\left[I\right]}=A_{\left[0\right]}\times \left(1-\frac{\left[I\right]}{\left[I\right]+{\mathrm{IC}}_{50}}\right)$$
where, *A*_[I]_, and *A*_[0]_ is the initial activity of DprE1 at inhibitor concentration [I] and the initial activity in the absence of inhibitor, respectively.

### 2.5. Metabolic Labeling of M. tuberculosis H37Rv

*M. tuberculosis* were incubated in 7H9 media supplemented with Tween 80 and ADC at 37 °C with shaking until OD_600_ 0.9. Radiolabeling with [^14^C]-acetate [specific activity: 110 mCi/mmol, American Radiolabeled Chemicals, Inc.] in the final concentration of 1 μCi/mL was performed as described [8], for 24 h, in the volume of 100 μL at concentrations 10 × MIC, 100 × MIC of the target compounds. The lipids were extracted with 1.5 mL of CHCl_3_/CH_3_OH (2:1) by incubation at 65 °C for 3 h. Following the biphasic Folch wash (2×), the samples were analyzed on TLC Silica gel 60 F_254_ plates in the solvent CHCl_3_/CH_3_OH/H_2_O (20:4:0.5, *v*/*v*) and the radiolabeled lipids were visualized using Amersham Typhoon 5 phosphorimager (GE Healthcare).

### 2.6. Liver Microsomal Metabolic Stability Test

The compounds’ metabolic stability were determined in human liver microsomes. A solution of microsomes (0.2 mg/mL) containing the compounds (1.0 mM), NADPH (1 mM), and phosphate buffer (100 mM, pH 7.4) were prepared. Pre-incubation was conducted at 37 °C for 10 min. There was addition of 50 µL and 5 mM NADPH to initiate the reaction. At 5, 15, 30, 60, or 120 min, 30 µL of reaction solution was taken out and quenched with 300 µL internal standard tolbutamide (10 ng/mL) in cold CH_3_CN. The mixture was centrifuged at 6000 rpm for 15 min at 4 °C. Then, 100 μL supernatant was diluted with 100 uL ultrapure water (Millipore, ZMQS50F01). The solution was analyzed via LC/MS analysis.

### 2.7. Pharmacokinetic Study in Mice

Animal Care and Welfare Committee of Shanghai Institute of Materia Medica, Chinese Academy of Sciences approved all animal protocols. All animal programs are in compliance with the Guide for the Care and Use of Laboratory Animals issued by Shanghai Association on Laboratory Animal Care (SALAC). SPF male ICR mice weighing 25–27 g were divided into two groups, three mice in each group. The tested compounds were administrated via oral and intravenous injection, separately. The compound was orally dosed at 5 mg/kg at a concentration as 1.0 mg/mL and intravenously (i.v.) dosed at 2 mg/kg as 0.4 mg/mL. The compound was co-dissolved by using 0.5% carboxymethyl cellulose for p.o. administration, and 10% DMSO/40%PEG400/40% water was used for i.v. administration, respectively. After oral dosing or i.v. administration, the blood was sampled at 5, 15, 30 min, and 1, 2, 4, 7, and 24 h. The sample plasma was harvested and stored at −80 °C for analysis. Based on noncompartmental analysis (Pharsight Corporation, Sunnyvale, CA, USA), the pharmacokinetic data were analyzed using WinNonlin software version 6.3.

### 2.8. In Vivo Efficacy Study

Animal work was approved by the Bioethics Committee at the Research Center of Biotechnology of RAS (Protocol N◦22/1, 11 February 2022) and was carried out according to the corresponding guidelines for animal use. Female 2-month-old BALB/c mice were kept in cages with a floor area of 960 cm^2^ and a height of 12 cm (North Kent Plastic Cages, Coalville, UK), in a special room with separate supply and exhaust ventilation without recirculation, equipped with a HEPA filter at the outlet, under standard conditions (natural light/dark cycle, air temperature 21–22 °C, relative humidity 50%, and 15–fold air exchange rate). Water and standard rodent compound feed (PK-120; Laboratorkorm, Moscow, Russia) were provided at libitum.

After 6 days of adaptation, mice were randomized into groups (*n* = 6) and inoculated with 0.2 µL of a suspension containing 3.6 × 10^6^ CFU of virulent *M. tuberculosis* strain H37Rv in PBS containing 0.0027 M KCl and 0.138 M NaCl per mouse intravenously in the lateral tail vein using a sterile 0.5 mL tuberculin syringe with a 28 G 1/2” needle (ISO-Med, Inc., Corona, CA, USA).

Solutions of test compounds were prepared in PBS buffer (pH 7.0, 50 mM) containing PEG-400 (40%) as a solubilizer and dispersed in an ultrasonic bath. One week after *Mtb* infection, the compounds were administered once a day intragastrically using a syringe pump (Harvard Apparatus, Holliston, MA, USA). Treatment was continued for 4 weeks, except weekends. Euthanasia was performed by dislocation of the cervical vertebrae.

To determine the number of Mtb CFU, the lungs from animals of each group were homogenized using a YellowlineD1 25 basic homogenizer (IKA-WERKE, Staufen im Breisgau, Germany). Ten-fold serial dilutions of the initial suspension in saline was prepared, 100 μL of each dilution was placed on a Petri dish with Dubos agar, and incubated for 21 days. The number of colonies per plate was then counted and the number of CFU of Mtb/lung was determined. The obtained quantitative data CFU Mtb/lung were converted to log10.

Statistical analysis was performed using an MS Office Excel software (Microsoft, version 2019, Redmond, WA, USA). Mean value (M), standard deviation (SD), and statistical significance (*p*) were evaluated at 95% confidence interval via Student’s *t*-test.

## 3. Results

### 3.1. Characterization Data for the Compounds

#### BTZ: 8-nitro-6-(trifluoromethyl)-4H-benzo[e][1,3]thiazin-4-one

*2-{[(1-cyclohexyl-1H-1,2,3-triazol-4-yl)methyl](methyl)amino}-BTZ* (**1**). Yield 53%; solid, yellow color; ^1^H NMR (400 MHz, CDCl_3_) *δ:* 9.13 (s, 1H), 8.77 (s, 1H), 7.77 (s, 1H), 5.12 (s, 2H), 4.43–4.37 (m, 1H), 3.51 (s, 3H), 2.16 (d, *J* = 8.0 Hz, 2H), 1.90 (d, *J* = 8.0 Hz, 2H), 1.76–1.70 (m, 2H), 1.47–1.38 (m, 2H), 1.30–1.25 (m, 2H); ^13^C NMR (151 MHz, CDCl_3_) *δ:* 163.3, 166.1, 141.4, 143.9, 133.7 (d, *J* = 2.2 Hz), 134.4, 130.0 (q, *J* = 35.6 Hz), 126.0 (d, *J* = 2.6 Hz), 126.7, 122.5 (q, *J* = 273.3 Hz), 121.6, 46.7, 60.5, 37.0, 25.3, 33.6, 25.2; HRMS-ESI (*m*/*z*) calcd [M+H]^+^ for C_19_H_20_F_3_N_6_O_3_S^+^ 469.1264, found 469.1259.

*2-{methyl[(1-phenyl-1H-1,2,3-triazol-4-yl)methyl]amino}-BTZ* (**2**). Yield 43%; solid, white; ^1^H NMR (400 MHz, CDCl_3_) *δ:* 9.13 (s, 1H), 8.76 (s, 1H), 8.24 (s, 1H), 7.70 (d, *J* = 7.60 Hz, 2H), 7.51 (t, *J* = 7.2 Hz, 2H), 7.44 (d, *J* = 7.2 Hz, 1H), 5.20 (s, 2H), 3.57 (s, 3H); ^13^C NMR (151 MHz, CDCl_3_) *δ:* 166.1, 163.4, 144.1, 142.5, 136.8, 134.2, 133.6 (d, *J* = 3.0 Hz), 130.0 (q, *J* = 36.1 Hz), 123.0, 129.0, 126.7, 126.1 (d, *J* = 3.0 Hz), 122.4 (q, *J* = 273.2 Hz), 122.3, 120.6, 46.7, 37.0; HRMS-ESI (*m*/*z*) calcd [M+H]^+^ for C_19_H_14_F_3_N_6_O_3_S^+^ 463.0795, found 463.0797.

*2-{ethyl[(1-phenyl-1H-1,2,3-triazol-4-yl)methyl]amino}-BTZ* (**3**). Yield 43%; solid, white; ^1^H NMR (400 MHz, CDCl_3_) *δ:* 9.12 (s, 1H), 8.76 (s, 1H), 8.33 (s, 1H), 7.71 (d, *J* = 8.0 Hz, 2H), 7.50 (t, *J* = 7.4 Hz, 2H), 7.43 (d, *J* = 6.8 Hz, 1H), 5.13 (s, 2H), 3.98 (q, *J* = 7.0 Hz, 2H), 1.50 (t, *J* = 3.8 Hz, 3H); ^13^C NMR (151 MHz, CDCl_3_) δ: 166.2, 162.6, 144.1, 142.9, 137.0, 134.5, 133.6, 129.9, 129.9 (q, *J* = 34.7 Hz), 129.1, 126.8, 126.2 (d, *J* = 3.0 Hz), 122.9, 122.5 (q, *J* = 273.3 Hz), 120.7, 45.1, 44.7, 13.1; HRMS-ESI (*m*/*z*) calcd [M+H]^+^ for C_20_H_16_F_3_N_6_O_3_S^+^ 477.0951, found 477.0944.

*2-{[(1-benzyl-1H-1,2,3-triazol-4-yl)methyl](methyl)amino}-BTZ* (**4**). Yield 40%; a white solid; ^1^H NMR (400 MHz, CDCl_3_) *δ:* 9.12 (s, 1H), 8.78 (s, 1H), 7.72 (s, 1H), 7.36 (br, 3H), 7.28 (br, 2H), 5.50 (s, 2H), 5.11 (s, 2H), 3.51 (s, 3H); ^13^C NMR (151 MHz, CDCl_3_) *δ:* 166.1, 163.4, 144.0, 142.3, 134.4, 134.4, 133.6 (d, *J* = 3.1 Hz), 130.0 (q, *J* = 36.1 Hz), 129.3, 129.0, 128.3, 126.7, 126.1 (d, *J* = 3.0 Hz), 123.9, 122.5 (q, *J* = 273.3 Hz), 54.3, 46.3, 36.8; HRMS-ESI (*m*/*z*) calcd [M+H]^+^ for C_20_H_16_F_3_N_6_O_3_S^+^ 477.0951, found 477.0944.

*2-{methyl[(1-(pyridin-4-yl)-1H-1,2,3-triazol-4-yl](methyl)amino}-BTZ* (**5**). Yield 49%; a yellow solid; ^1^H NMR (400 MHz, DMSO-*d_6_*) *δ:* 9.00 (s, 1H), 8.88 (s, 2H), 8.82 (br, 2H), 7.98 (s, 2H), 5.21 (s, 2H), 3.44 (s, 3H); ^13^C NMR (151 MHz, CDCl_3_) *δ:* 166.1, 163.6, 151.7, 144.0, 143.4, 142.6, 134.3, 133.7 (d, *J* = 2.9 Hz), 130.1 (q, *J* = 35.9 Hz), 126.7 (d, *J* = 3.3 Hz), 126.3 (d, *J* = 3.3 Hz), 122.5 (q, *J* = 273.2 Hz), 121.9, 46.8, 37.2; HRMS-ESI (*m*/*z*) calcd [M+H]^+^ for C_18_H_13_F_3_N_7_O_3_S^+^ calculated 464.0747, found 464.0748.

*2-{methyl[(1-(pyridin-3-yl)-1H-1,2,3-triazol-4-yl](methyl)amino}-BTZ* (**6**). Yield 57%; a yellow solid; ^1^H NMR (400 MHz, CDCl_3_) *δ:* 9.12 (s, 1H), 9.04 (s, 1H), 8.77 (s, 1H), 8.72 (brs, 1H), 8.35 (brs, 1H), 8.08 (d, *J* = 7.4 Hz, 1H), 7.50 (brs, 1H), 5.20 (s, 2H), 3.58 (s, 3H); ^13^C NMR (151 MHz, CDCl_3_) *δ:* 166.2, 163.6, 150.3, 150.2, 144.0, 143.2, 141.9, 134.3, 133.5 (d, *J* = 2.6 Hz), 130.1 (q, *J* = 35.5 Hz), 128.1, 126.6, 126.2 (d, *J* = 2.9 Hz), 122.5, 122.5 (q, *J* = 273.3 Hz), 121.6, 46.7, 37.3; HRMS-ESI (*m*/*z*) calcd [M+Na]^+^ for C_18_H_12_F_3_N_7_NaO_3_S^+^ 486.0567, found 486.0547.

*2-{[(1-(1,1′-biphenyl)-4-yl-1H-1,2,3-triazol-4-yl](methyl)(methyl)amino}-BTZ* (**7**) Yield 64%; a yellow solid; ^1^H NMR (400 MHz, CDCl_3_) *δ:* 9.14 (s, 1H), 8.77 (s, 1H), 8.30 (s, 1H), 7.79 (d, *J* = 8.0 Hz, 2H), 7.72 (d, *J* = 8.0 Hz, 2H), 7.60 (d, *J* = 7.2 Hz, 2H), 7.47 (t, *J* = 7.4 Hz, 2H), 7.39 (t, *J* = 7.2 Hz, 1H), 5.21 (s, 2H), 3.59 (s, 3H); ^13^C NMR (151 MHz, CDCl_3_) *δ:* 166.2, 163.5, 144.0, 142.6, 142.1, 139.6, 136.0, 134.4, 133.7, 130.0 (q, *J* = 35.49 Hz), 129.1, 128.5, 128.2, 127.2, 126.7, 126.2 (d, *J* = 2.6 Hz), 122.6 (q, *J* = 273.1 Hz), 122.2, 120.9, 46.8, 37.0; HRMS-ESI (*m*/*z*) calcd [M+H]^+^ for C_25_H_18_F_3_N_6_O_3_S^+^ 539.1108, found 539.1108.

*2-{methyl[(1-(naphthalen-2-yl)-1H-1,2,3-triazol-4-yl](methyl)amino}-BTZ* (**8**). Yield 67%; yellow solid; ^1^H NMR (400 MHz, CDCl_3_) *δ:* 9.14 (d, *J* = 1.6 Hz, 1H), 8.77 (d, *J* = 1.6 Hz, 1H), 8.39 (s, 1H), 8.16 (d, *J* = 1.6 Hz, 1H), 7.99 (d, *J* = 8.8 Hz, 1H), 7.93–7.86 (m, 3H), 7.60–7.54 (m, 2H), 5.24 (s, 2H), 3.60 (s, 3H); ^13^C NMR (151 MHz, CDCl_3_) *δ:* 166.2 163.6, 134.4, 134.3, 133.6 (d, *J* = 3.4 Hz), 133.3, 133.1, 130.2, 129.9 (q, *J* = 35.48 Hz), 128.4, 128.1, 127.7, 127.3, 126.6, 126.2 (d, *J* = 2.6 Hz), 122.5 (q, *J* = 273.1 Hz), 122.4, 118.9, 118.7, 46.7, 37.1; HRMS-ESI (*m*/*z*) calcd [M+H]^+^ for C_23_H_16_F_3_N_6_O_3_S^+^ 513.0951, found 513.0949.

*2-{methyl[(1-(naphthalen-1-yl)-1H-1,2,3-triazol-4-yl](methyl)amino}-BTZ* (**9**). Yield 60%; solid, yellow color; ^1^H NMR (400 MHz, CDCl_3_) *δ:* 9.10 (s, 1H), 8.77 (d, *J* = 1.6 Hz, 1H), 8.19 (s, 1H), 8.02 (d, *J* = 8.2 Hz, 1H), 7.95 (d, *J* = 8.0 Hz, 1H), 7.60–7.51 (m, 5H), 5.27 (s, 2H), 3.65 (s, 3H); ^13^C NMR (151 MHz, CDCl_3_) *δ:* 166.1, 163.5, 144.0, 141.9, 134.3, 133.7, 133.5, 130.7, 130.0 (q, *J* = 35.4 Hz), 128.5, 128.4, 128.1, 127.3, 126.7, 126.2, 125.1, 123.6, 122.5 (q, *J* = 273.1 Hz), 122.3, 46.8, 37.2; HRMS-ESI (*m*/*z*) calcd [M+H]^+^ for C_23_H_16_F_3_N_6_O_3_S^+^ 513.0951, found 513.0951.

*2-{methyl[(1-(thiophen-2-ylmethyl)-1H-1,2,3-triazol-4-yl](methyl)amino}-BTZ* (**10**). Yield 71%; solid, yellow color; ^1^H NMR (400 MHz, CDCl_3_) *δ:* 9.11 (d, *J* = 1.6 Hz, 1H), 8.76 (d, *J* = 1.6 Hz, 1H), 7.77 (s, 1H), 7.31 (d, *J* = 4.8 Hz, 1H), 7.10 (d, *J* = 2.8 Hz, 1H), 7.03–6.93 (m, 1H), 5.67 (s, 2H), 5.10 (s, 2H), 3.49 (s, 3H); ^13^C NMR (151 MHz, CDCl_3_) *δ:* 166.1, 163.5, 144.0, 142.2, 135.8, 134.4, 133.6 (d, *J* = 2.6 Hz), 129.9 (q, *J* = 35.5 Hz), 128.5, 127.5, 127.4, 126.7, 126.1 (d, *J* = 2.8 Hz), 123.6, 122.5 (q, *J* = 273.3 Hz), 48.7, 46.5, 36.9; HRMS-ESI (*m*/*z*) calcd [M+H]^+^ for C_18_H_14_F_3_N_6_O_3_S_2_^+^ 483.0515, found 483.0512.

*2-{methyl[(1-(thiazol-4-ylmethyl)-1H-1,2,3-triazol-4-yl](methyl)amino}-BTZ* (**11**). Yield 43%; solid, yellow color; ^1^H NMR (400 MHz, CDCl_3_) *δ:* 9.11 (d, *J* = 2.0 Hz, 1H), 8.80 (s, 1H), 8.76 (d, *J* = 2.0 Hz, 1H), 7.91 (s, 1H), 7.32 (s, 1H), 5.68 (s, 2H), 5.12 (s, 2H), 3.49 (s, 3H); ^13^C NMR (151 MHz, CDCl_3_) *δ:* 166.1, 163.5, 154.3, 150.2, 144.0, 142.2, 134.3, 133.6 (d, *J* = 3.3 Hz), 129.9 (q, *J* = 35.6 Hz), 126.7, 126.1 (d, *J* = 3.3 Hz), 124.3, 122.5 (d, *J* = 273.1 Hz), 118.1, 49.8, 46.5, 36.9; HRMS-ESI (*m*/*z*) calcd [M+H]^+^ for C_17_H_13_F_3_N_7_O_3_S_2_^+^ 484.0468, found 484.0467.

*2-{[(1-(4-fluorophenyl)-1H-1,2,3-triazol-4-yl](methyl)(methyl)amino}-BTZ* (**12**). Yield 66%; solid, yellow color; ^1^H NMR (400 MHz, CDCl_3_) *δ:* 9.12 (s, 1H), 8.78 (s, 1H), 8.26 (s, 1H), 7.69 (dd, *J* = 7.6, 4.4 Hz, 2H), 7.19 (t, *J* = 8.0 Hz, 2H), 5.20 (s, 2H), 3.57 (s, 3H); ^13^C NMR (151 MHz, CDCl_3_) *δ:* 166.2, 163.6, 162.8 (d, *J* = 250.0 Hz), 143.9, 142.7, 134.3, 133.6 (d, *J* = 2.4 Hz), 133.1, 130.0 (q, *J* = 35.1 Hz), 126.7, 126.3 (d, *J* = 3.0 Hz), 122.7 (d, *J* = 8.4 Hz), 122.6, 122.5 (q, *J* = 273.1 Hz), 116.9 (d, *J* = 23.3 Hz), 46.6, 37.1; HRMS-ESI (*m*/*z*) calcd [M+H]^+^ for C_19_H_13_F_4_N_6_O_3_S^+^ 481.0700, found 481.0692.

*2-{[(1-(3-fluorophenyl)-1H-1,2,3-triazol-4-yl](methyl)(methyl)amino}-BTZ* (**13**). Yield 59%; solid, yellow color ^1^H NMR (400 MHz, CDCl_3_) *δ:* 9.13 (s, 1H), 8.78 (s, 1H), 8.30 (s, 1H), 7.56–7.47 (m, 3H), 7.14 (s, 1H), 5.18 (s, 2H), 3.58 (s, 3H); ^13^C NMR (151 MHz, CDCl_3_) *δ:* 166.2, 163.6, 163.2 (d, *J* = 249.0 Hz), 143.9, 142.8, 138.0 (d, *J* = 9.5 Hz), 134.3, 133.7 (d, *J* = 2.4 Hz), 131.4 (d, *J* = 8.5 Hz), 132.9 (q, *J* = 35.5 Hz), 126.6, 126.2 (d, *J* = 2.8 Hz), 122.5 (q, *J* = 273.0 Hz), 122.4, 116.1, 116.0, 108.5 (d, *J* = 26.3 Hz), 46.7, 37.2; HRMS-ESI (*m*/*z*) calcd [M+H]^+^ for C_19_H_13_F_4_N_6_O_3_S^+^ 481.0700, found 481.0701.

*2-{[(1-(2-fluorophenyl)-1H-1,2,3-triazol-4-yl](methyl)(methyl)amino}-BTZ* (**14**). Yield 53%; solid, yellow color; ^1^H NMR (400 MHz, CDCl_3_) *δ:* 9.12 (s, 1H), 8.76 (s, 1H), 8.29 (s, 1H), 7.88 (br, 1H), 7.44 (d, *J* = 5.2 Hz, 1H), 7.31 (t, *J* = 8.2 Hz, 2H), 5.23 (s, 2H), 3.56 (s, 3H); ^13^C NMR (151 MHz, CDCl_3_) *δ:* 166.1, 163.6, 153.5 (d, *J* = 251.8 Hz), 143.9, 142.3, 134.2, 133.8 (d, *J* = 2.9 Hz), 130.7 (d, *J* = 6.8 Hz), 130.0 (q, *J* = 35.4 Hz), 126.7, 126.1, 125.4, 125.3, 125.1, 125.0, 122.5 (q, *J* = 273.2 Hz), 117.3 (d, *J* = 19.9 Hz), 46.5, 37.1; HRMS-ESI (*m*/*z*) calcd [M+H]^+^ for C_19_H_13_F_4_N_6_O_3_S^+^ 481.0700, found 481.0696.

*2-{[(1-(3-chlorophenyl)-1H-1,2,3-triazol-4-yl](methyl)(methyl)amino}-BTZ* (**15**). Yield 49%; solid, white color; ^1^H NMR (400 MHz, CDCl_3_) *δ:* 9.14 (s, 1H), 8.78 (s, 1H), 8.25 (s, 1H), 7.77 (s, 1H), 7.60 (d, *J* = 7.3 Hz, 1H), 7.47–7.42 (m, 2H), 5.20 (s, 2H), 3.57 (s, 3H); ^13^C NMR (151 MHz, CDCl_3_) *δ:* 166.2, 163.6, 144.0, 142.9, 137.7, 135.8, 134.3, 133.7, 131.0, 130.0 (q, *J* = 35.9 Hz), 129.2, 126.7, 126.2, 122.6, 122.5 (q, *J* = 273.4 Hz), 121.0, 118.7, 46.8, 37.4; HRMS-ESI (*m*/*z*) calcd [M+H]^+^ for: C_19_H_13_ClF_3_N_6_O_3_S^+^ 497.0405, found 497.0407.

*2-{[(1-(3-bromophenyl)-1H-1,2,3-triazol-4-yl](methyl)(methyl)amino}-BTZ* (**16**). Yield 57%; solid, white color; ^1^H NMR (400 MHz, CDCl_3_) *δ:* 9.13 (s, 1H), 8.77 (s,1H), 8.26 (s, 1H), 7.93 (s, 1H), 7.65 (d, *J* = 7.6 Hz, 1H), 7.56 (d, *J* = 7.6 Hz, 1H), 7.38 (t, *J* = 7.8 Hz, 1H), 5.19 (s, 2H), 3.57 (s, 3H); ^13^C NMR (151 MHz, CDCl_3_) *δ:* 166.2, 163.4, 144.0, 142.9, 137.8, 134.2, 133.6 (d, *J* = 2.4 Hz), 132.2, 131.2, 130.0 (q, *J* = 35.8 Hz), 126.5, 126.1 (d, *J* = 2.7 Hz), 123.8, 123.5, 122.5 (q, *J* = 273.0 Hz), 122.2, 119.1, 46.6, 37.2; HRMS-ESI (*m*/*z*) calcd [M+H]^+^ for C_19_H_13_BrF_3_N_6_O_3_S^+^ 540.9900; 542.9879, found 540.9899, 542.9880.

*2-{[(1-(3-iodophenyl)-1H-1,2,3-triazol-4-yl](methyl)(methyl)amino}-BTZ* (**17**). Yield 70%; solid, white color. ^1^H NMR (400 MHz, CDCl_3_) *δ:* 9.13 (s, 1H), 8.77 (s, 1H), 8.24 (s, 1H), 8.10 (s, 1H), 7.76 (d, *J* = 7.6 Hz, 1H), 7.68 (d, *J* = 7.6 Hz, 1H), 7.22 (t, *J* = 7.6 Hz, 1H), 5.19 (s, 2H), 3.57 (s, 3H); ^13^C NMR (151 MHz, CDCl_3_) *δ:* 166.1, 163.6, 144.1, 142.8, 138.2, 137.6, 134.3, 133.7, 131.3, 130.0 (q, *J* = 35.2 Hz), 129.5, 126.6, 126.2 (d, *J* = 2.6 Hz), 122.5 (q, *J* = 273.2 Hz), 122.2, 119.8, 94.6, 46.6, 37.1; HRMS-ESI (*m*/*z*) calcd [M+H]^+^ for C_19_H_13_F_3_IN_6_O_3_S^+^ 588.9761, found: 588.9760.

*2-{[(1-(3-methoxyphenyl)-1H-1,2,3-triazol-4-yl](methyl)(methyl)amino}-BTZ* (**18**). Yield 57%; solid, white color; ^1^H NMR (400 MHz, CDCl_3_) *δ:* 9.12 (s, 1H), 8.77 (s, 1H), 8.24 (s, 1H), 7.39 (t, *J* = 8.0 Hz, 1H), 7.30 (s, 1H), 7.23 (d, *J* = 7.8 Hz, 1H), 6.95 (d, *J* = 7.8 Hz, 1H), 5.20 (s, 2H), 3.86 (s, 3H), 3.57 (s, 3H); ^13^C NMR (151 MHz, CDCl_3_) *δ:* 166.1, 163.5, 160.8, 144.1, 142.5, 137.9, 134.3, 133.7, 130.7, 130.0 (q, *J* = 35.6 Hz), 126.7, 126.2 (d, *J* = 2.6 Hz), 122.5 (q, *J* = 273.2 Hz), 122.3, 115.0, 112.6, 106.5, 55.8, 46.6, 37.1; HRMS-ESI (*m*/*z*) calcd [M+H]^+^ for C_20_H_16_F_3_N_6_O_4_S^+^ 493.0900, found 493.0900.

*2-{methyl[1-(m-tolyl)-1H-1,2,3-triazol-4-yl](methyl)amino}-BTZ* (**19**). Yield 66%; solid, white color; ^1^H NMR (400 MHz, CDCl_3_) *δ:* 9.13 (s, 1H), 8.77 (s, 1H), 8.22 (s, 1H), 7.53 (s, 1H), 7.48 (d, *J* = 8.0 Hz, 1H), 7.37 (t, *J* = 7.6 Hz, 1H), 7.23 (d, *J* = 7.6 Hz, 1H), 5.20 (s, 2H), 3.57 (s, 3H), 2.43 (s, 3H); ^13^C NMR (151 MHz, CDCl_3_) *δ:* 166.2, 163.5, 144.0, 142.5, 140.2, 136.8, 134.4, 133.7 (d, *J* = 2.5 Hz), 130.0 (q, *J* = 35.6 Hz), 129.9, 129.7, 126.7, 126.2 (d, *J* = 2.7 Hz), 122.5 (q, *J* = 273.2 Hz), 122.3, 121.3, 117.8, 46.6, 37.1, 21.5; HRMS-ESI (*m*/*z*) calcd [M+H]^+^ for C_20_H_16_F_3_N_6_O_3_S^+^ 477.0951, found 477.0948.

*2-{methyl[(1-3-(trifluoromethyl)phenyl]-1H-1,2,3-triazol-4-yl)(methyl)amino-BTZ* (**20**). Yield 65%; solid, white color, ^1^H NMR (400 MHz, CDCl_3_) *δ:* 9.13 (s, 1H), 8.79 (s, 1H), 8.34 (s, 1H), 8.04 (br, 1H), 7.91 (br, 1H), 7.68 (d, *J* = 12.0 Hz, 2H), 5.20 (s, 2H), 3.58 (s, 3H); ^13^C NMR (151 MHz, CDCl_3_) *δ:* 166.2, 163.7, 144.0, 143.1, 137.2, 134.3, 132.7 (q, *J* = 33.5 Hz), 130.7, 130.0 (q, *J* = 35.6 Hz), 129.6 (d, *J* = 4.0 Hz), 126.6, 126.2 (d, *J* = 3.3 Hz), 125.8, 123.8 (q, *J* = 272.8 Hz), 123.7, 123.4 (q, *J* = 272.7 Hz), 122.4, 117.8 (d, *J* = 3.7 Hz), 46.6, 37.2; HRMS-ESI (*m*/*z*) calcd [M+H]^+^ for C_20_H_13_F_6_N_6_O_3_S^+^ 531.0669, found 531.0669.

*3-{4-[(methyl(8-nitro-4-oxo-6-(trifluoromethyl)-4H-benzo[e]*[1,3]*thiazin-2-yl)amino)methyl)-1H-1,2,3-triazol-1-yl)benzonitrile* (**21**). Yield 63%; solid, white color. ^1^H NMR (400 MHz, CDCl_3_) δ: 9.13 (s, 1H), 8.77 (s, 1H), 8.36 (s, 1H), 8.10 (s, 1H), 7.98 (d, *J* = 7.60 Hz, 1H), 7.73 (d, *J* = 7.40 Hz, 1H), 7.66 (t, *J* = 7.80 Hz, 1H), 5.19 (s, 2H), 3.59 (s, 3H); ^13^C NMR (151 MHz, CDCl_3_) δ: 166.1, 163.5, 143.9, 143.3, 137.5, 134.3, 133.7 (d, *J* = 2.30 Hz), 132.4, 131.2, 130.1 (q, *J* = 35.6 Hz), 126.6, 126.3 (d, *J* = 3.1 Hz), 124.5, 123.9, 122.5 (q, *J* = 273.3 Hz), 122.3, 117.4, 114.5, 46.7, 37.3; HRMS-ESI (*m*/*z*) calcd [M+H]^+^ for C_20_H_13_F_3_N_7_O_3_S^+^ 488.0747, found 488.074.

*2-{[(1-(3,4-difluorophenyl)-1H-1,2,3-triazol-4-yl](methyl)(methyl)amino}-BTZ* (**22**). Yield 61%; solid, white color; ^1^H NMR (400 MHz, CDCl_3_) *δ:* 9.13 (s, 1H), 8.78 (s, 1H), 8.26 (s, 1H), 7.68–7.64 (m, 1H), 7.44 (brs, 1H), 7.35–7.28 (m, 1H), 5.17 (s, 2H), 3.58 (s, 3H); ^13^C NMR (151 MHz, CDCl_3_) *δ:* 166.2, 163.6, 150.8 (dd, *J* = 252.2, 13.60 Hz), 150.5 (dd, *J* = 253.70, 13.6 Hz), 144.0, 143.0, 134.3 133.7 (d, *J* = 1.50 Hz),133.2, 130.1 (q, *J* = 36.24 Hz), 126.6, 126.2 (d, *J* = 3.0 Hz), 122.5 (q, *J* = 273.3 Hz), 122.5, 118.6 (d, *J* = 19.4 Hz), 116.7, 110.7 (d, *J* = 22.7 Hz), 46.7, 37.3; HRMS-ESI (*m*/*z*) calcd [M+H]^+^ for C_19_H_12_F_5_N_6_O_3_S^+^ 499.0606, found 499.0606.

*2-{[(1-(3,5-difluorophenyl)-1H-1,2,3-triazol-4-yl](methyl)(methyl)amino}-BTZ* (**23**) Yield 43%; solid, white color. ^1^H NMR (400 MHz, CDCl_3_) *δ:* 9.13 (s, 1H), 8.78 (s, 1H), 8.30 (s, 1H), 7.34 (d, *J* = 5.20 Hz, 2H), 6.89 (brs, 1H), 5.18 (s, 2H), 3.58 (s, 3H); ^13^C NMR (151 MHz, CDCl_3_) *δ:* 166.2, 163.6, 163.55 (dd, *J* = 251.10, 13.7 Hz), 144.0, 143.1, 138.4 (d, *J* = 8.90 Hz), 134.3, 133.7 (d, *J* = 2.60 Hz), 130.1 (q, *J* = 35.40 Hz), 126.6, 126.3 (d, *J* = 3.0 Hz), 122.5 (q, *J* = 273.30 Hz), 122.4, 104.3 (d, *J* = 40.10 Hz), 104.2 (d, *J* = 30.50 Hz), 46.7, 37.3; HRMS-ESI (*m*/*z*) calcd [M+H]^+^ for C_19_H_12_F_5_N_6_O_3_S^+^ 499.0606, found 499.0603.

*2-{[(1-(2,6-difluorophenyl)-1H-1,2,3-triazol-4-yl](methyl)(methyl)amino}-BTZ* (**24**). Yield 57%; solid, white color; ^1^H NMR (400 MHz, CDCl_3_) *δ:* 9.11 (s, 1H), 8.77 (s, 1H), 8.10 (s, 1H), 7.50–7.47 (m, 1H), 7.12 (t, *J* = 8.4 Hz, 2H), 5.22 (s, 2H), 3.59 (s, 3H); ^13^C NMR (151 MHz, CDCl_3_) *δ:* 166.1, 163.6, 156.9 (d, *J* = 256.70 Hz), 144.0, 141.8, 134.4, 133.7 (d, *J* = 2.10 Hz), 131.71 (t, *J* = 9.10 Hz), 130.0 (q, *J* = 35.50 Hz), 127.0, 126.7, 126.2 (d, *J* = 3.00 Hz), 122.5 (q, *J* = 273.30 Hz), 115.1 (t, *J* = 15.10 Hz), 112.7 (d, *J* = 19.80 Hz), 46.6, 37.2; HRMS-ESI (*m*/*z*) calcd [M+H]^+^ for C_19_H_12_F_5_N_6_O_3_S^+^ 499.0606, found 499.0602.

*2-{[(1-(2,4-difluorophenyl)-1H-1,2,3-triazol-4-yl](methyl)(methyl)amino}-BTZ* (**25**). Yield 85%; solid, white color^1^H NMR (400 MHz, CDCl_3_) *δ:* 9.13 (s, 1H), 8.78 (s, 1H), 8.26 (s, 1H), 7.88–7.82 (m, 1H), 7.08– 7.03 (m, 2H), 5.21 (s, 2H), 3.58 (s, 3H); ^13^C NMR (151 MHz, CDCl_3_) *δ:* 166.0, 163.6, 162.7 (dd, *J* = 253.40, 10.9 Hz), 154.1 (dd, *J* = 254.70, 11.8 Hz), 144.0, 142.4, 134.3, 133.7 (d, *J* = 2.80 Hz), 130.0 (q, *J* = 35.50 Hz), 126.7, 126.4 (d, *J* = 9.90 Hz), 126.1 (d, *J* = 2.60 Hz), 125.3 (d, *J* = 4.10 Hz), 122.5 (q, *J* = 273.60 Hz), 121.9 (d, *J* = 7.20 Hz), 112.7 (d, *J* = 22.40 Hz), 105.6 (dd, *J* = 78.5, 2.30 Hz), 46.5, 37.1; HRMS-ESI (*m*/*z*) calcd [M+H]^+^ for C_19_H_12_F_5_N_6_O_3_S^+^ 499.0606, found 499.0607.

*2-{[(1-(3,4-dichlorophenyl)-1H-1,2,3-triazol-4-yl](methyl)(methyl)amino}-BTZ* (**26**). Yield 65%; solid, white color. ^1^H NMR (400 MHz, CDCl_3_) *δ:* 9.14 (s, 1H), 8.78 (s, 1H), 8.29 (s, 1H), 7.91 (s, 1H), 7.59 (br, 2H), 5.18 (s, 2H), 3.57 (s, 3H); ^13^C NMR (151 MHz, CDCl_3_) δ: 166.3, 163.6, 144.1, 143.1, 135.9, 134.2, 133.6 (d, *J* = 2.000 Hz), 133.4, 131.6, 130.1 (q, *J* = 35.3 Hz), 126.5, 126.3 (d, *J* = 3.0 Hz), 122.5 (q, *J* = 273.30 Hz), 122.5, 122.4, 122.3, 119.6, 46.7, 37.2; HRMS-ESI (*m*/*z*) calcd [M+H]^+^ for C_19_H_12_Cl_2_F_3_N_6_O_3_S^+^ 531.0015, found 531.0012.

*2-{[(1-(3-bromo-4-fluorophenyl)-1H-1,2,3-triazol-4-yl](methyl)(methyl)amino}-BTZ* (**27**). Yield 74%; solid, white color. ^1^H NMR (400 MHz, CDCl_3_) δ: 9.13 (s, 1H), 8.78 (s, 1H), 8.25 (s, 1H), 7.98 (d, *J* = 3.2 Hz, 1H), 7.65 (d, *J* = 8.4 Hz, 1H), 7.29 (br, 1H), 5.18 (s, 2H), 3.58 (s, 3H); Poor solubility to obtain a ^13^C NMR. HRMS-ESI (*m*/*z*) calcd [M+H]^+^ for C_19_H_12_BrF_4_N_6_O_3_S^+^ 558.9806, 560.9785, found 558.9794,560.9766.

*2-{[(1-(3-chloro-4-fluorophenyl)-1H-1,2,3-triazol-4-yl](methyl)(methyl)amino}-BTZ* (**28**). Yield 87%; solid, white color. ^1^H NMR (400 MHz, CDCl_3_) *δ:* 9.13 (s, 1H), 8.78 (s, 1H), 8.24 (s, 1H), 7.85 (d, *J* = 4.41 Hz, 1H), 7.61 (d, *J* = 7.19 Hz, 1H), 7.31–7.29 (m, 1H), 5.17 (s, 2H), 3.56 (s, 3H); ^13^C NMR (151 MHz, CDCl_3_) δ: 166.3, 163.5, 158.2 (d, *J* = 252.2 Hz), 144.0, 143.0, 134.4, 133.6, 133.5, 130.1 (q, *J* = 34.7 Hz), 126.5, 126.3 (d, *J* = 3.0 Hz), 123.3, 122.8, 122.5, 122.5 (q, *J* = 273.3 Hz), 120.4 (d, *J* = 7.6 Hz), 117.8 (d, *J* = 22.7 Hz), 46.8, 37.3; HRMS-ESI (*m*/*z*) calcd [M+H]^+^ for C_19_H_12_ClF_4_N_6_O_3_S^+^ 515.0311, found 515.0303.

*2-{[(1-(2,4-dichlorophenyl)-1H-1,2,3-triazol-4-yl](methyl)(methyl)amino}-BTZ* (**29**). Yield 72%; solid, white color; ^1^H NMR (400 MHz, CDCl_3_) *δ:* 9.11 (d, *J* = 1.62 Hz, 1H), 8.77 (d, *J* = 1.62 Hz, 1H), 8.22 (s, 1H), 7.58 (d, *J* = 2.0 Hz, 1H), 7.52 (d, *J* = 8.8 Hz, 1H), 7.41 (dd, *J* = 8.4, 1.6 Hz, 1H), 5.21 (s, 2H), 3.58 (s, 3H); ^13^C NMR (151 MHz, CDCl_3_) *δ:* 166.0, 163.4, 143.9, 141.9, 136.6, 134.4, 133.8 (d, *J* = 2.6 Hz), 133.4, 130.8, 130.0 (q, *J* = 35.6 Hz), 129.6, 128.5, 128.4, 126.6, 126.1, 122.5 (q, *J* = 273.0 Hz), 46.6, 37.3; HRMS-ESI (*m*/*z*) calcd [M+H]^+^ for C_19_H_12_Cl_2_F_3_N_6_O_3_S^+^ 531.0015, found 531.0015.

*2-{[(1-(3,5-dichlorophenyl)-1H-1,2,3-triazol-4-yl](methyl)(methyl)amino}-BTZ* (**30**). Yield 75%; solid, white color. ^1^H NMR (400 MHz, CDCl_3_) *δ:* 9.13 (d, *J* = 1.6 Hz, 1H), 8.78 (d, *J* = 1.2 Hz, 1H), 8.28 (s,1H), 7.68 (d, *J* = 1.6 Hz, 2H), 7.42 (s, 1H), 5.20 (s, 2H), 3.56 (s, 3H); ^13^C NMR (151 MHz, CDCl_3_) *δ:* 166.2, 163.7, 144.0, 143.1, 138.1, 136.5, 134.2, 133.7 (d, *J* = 2.8 Hz), 130.1 (q, *J* = 35.2 Hz), 129.1, 126.6, 126.2 (d, *J* = 2.7 Hz), 122.6 (q, *J* = 273.0 Hz), 122.2, 119.1, 46.6, 37.2; HRMS-ESI (*m*/*z*) calcd [M+H]^+^ for C_19_H_12_Cl_2_F_3_N_6_O_3_S^+^ 531.0015, found 531.0016.

*2-{methyl[(2-phenyloxazol-4-yl)methyl]amino}-BTZ* (**31**). Solid, white color; ^1^H NMR indicates 3:1 atropisomeric ratio through the integral value of −CH_2_ protons and the oxazole—CH protons; ^1^H NMR (400 MHz, CDCl_3_) *δ:* 9.15 (s, 1H), 8.76 (s, 1H), 8.01 (br, 2H), 7.85 (s, 0.75H, major), 7.77 (s, 0.25H, minor), 7.45 (br, 3H), 5.06 (s, 1.5H, major), 4.88 (s, 0.5H, minor), 3.57 (s, 3H); ^13^C NMR (151 MHz, CDCl_3_) *δ:* 166.2, 166.1, 162.1, 144.0, 137.6, 136.2, 134.4, 133.7, 130.8, 129.9 (q, *J* = 35.40 Hz), 129.0, 126.8, 126.7, 126.1, 122.5 (q, *J* = 273.3 Hz), 116.3, 46.9, 37.1; HRMS-ESI (*m*/*z*) calcd [M+H]^+^ for C_20_H_14_F_3_N_4_O_4_S^+^ 463.0682, found 463.0683.

*2-{[(2-(4-fluorophenyl)oxazol-4-yl](methyl)(methyl)amino}-BTZ* (**32**). Solid, white; ^1^H NMR indicates 3:1 atropisomeric ratio through the integral value of −CH_2_ protons and the oxazole −CH protons; ^1^H NMR (400 MHz, CDCl_3_) *δ:* 9.09 (s, 1H), 8.72 (s, 1H), 7.96 (br, 2H), 7.78 (s, 0.75H, major), 7.72 (s, 0.25H, minor), 7.09 (t, *J* = 7.5 Hz, 2H), 4.99 (s, 1.5H, major), 4.81 (s, 0.5H, minor), 3.51 (s, 3H); ^13^C NMR (151 MHz, CDCl_3_) *δ:* 166.1, 164.3 (d, *J* = 252.4 Hz), 163.5, 161.2, 144.0, 137.6, 136.2, 134.4, 133.7, 129.9 (d, *J* = 35.6 Hz), 128.7 (d, *J* = 7.9 Hz), 126.7, 126.1, 123.6, 122.5 (d, *J* = 273.0 Hz), 116.0 (d, *J* = 22.1 Hz), 46.9, 37.1; HRMS-ESI (*m*/*z*) calcd [M+H]^+^ for C_20_H_13_F_4_N_4_O_4_S^+^ 481.0588, found 481.0586.

*2-{[(2-(3-fluorophenyl)oxazol-4-yl](methyl)(methyl)amino}-BTZ* (**33**). A white solid; ^1^H NMR indicates 3:1 atropisomeric ratio through the integral value of −CH_2_ protons and the oxazole −CH protons; ^1^H NMR (400 MHz, CDCl_3_) *δ:* 9.09 (s, 1H), 8.73 (s, 1H), 7.95 (s, 1H), 7.86 (s, 0.75H, major), 7.79 (s, 0.25H, minor), 7.41 (br, 1H), 7.21–7.10 (m, 2H), 5.04 (s, 1.5H, major), 4.85 (s, 0.5H, minor), 3.52 (s, 3H); ^13^C NMR (151 MHz, CDCl_3_) *δ:* 166.1, 163.5, 160.1 (d, *J* = 256.0 Hz), 158.4, 138.0, 137.6, 136.2, 134.4, 133.7, 132.4 (d, *J* = 7.2 Hz), 129.9 (d, *J* = 35.6 Hz), 129.7, 128.9, 126.7, 126.1, 124.5, 122.5 (q, *J* = 273.2 Hz), 117.0 (d, *J* = 21.2 Hz), 46.8, 37.1; HRMS-ESI (*m*/*z*) calcd [M+H]^+^ for C_20_H_13_F_4_N_4_O_4_S^+^ 481.0588, found 481.0587.

*2-{methyl[(2-phenylthiazol-4-yl)methyl]amino}-BTZ* (**34**). Solid, white color; ^1^H NMR indicates 3:1 atropisomeric ratio through the integral value of −CH_2_ protons; ^1^H NMR (400 MHz, CDCl_3_) *δ:* 9.13 (s, 1H), 8.76 (d, *J* = 1.9 Hz, 1H), 7.96–7.88 (m, 2H), 7.44–7.41 (m, 4H), 5.22 (s, 1.5H, major), 5.03 (s, 0.5H, minor), 3.57 (s, 3H); ^13^C NMR (151 MHz, CDCl_3_) δ: 168.7, 166.2, 163.5, 151.1, 144.0, 134.5, 133.7, 130.4, 129.8 (q, *J* = 35.4 Hz), 129.1, 126.9, 126.6, 126.1, 122.5 (q, *J* = 273.3 Hz), 118.2, 116.8, 50.7, 37.2; HRMS-ESI (*m*/*z*) calcd [M+H]^+^ for C_20_H_14_F_3_N_4_O_3_S_2_^+^ 479.0454, found 479.0451.

*2-{[(2-(3-fluorophenyl)thiazol-4-yl](methyl)(methyl)amino}-BTZ* (**35**). Solid, white color; ^1^H NMR indicates 3:1 atropisomeric ratio through the integral value of −CH_2_ protons and the thiazole −CH protons; ^1^H NMR (400 MHz, CDCl_3_) *δ:* 9.13 (s, 1H), 8.76 (s, 1H), 8.24 (br, 1H), 7.91 (s, 0.75H, major), 7.52 (s, 0.25H, minor), 7.42 (br, 2H), 7.23–7.16(m, 1H), 5.24 (s, 1.5H, major), 5.06 (s, 0.5H, minor), 3.57 (s, 3H); ^13^C NMR (151 MHz, CDCl_3_) *δ:* 166.2, 163.6, 160.1 (d, *J* = 253.5 Hz), 150.1, 144.0, 134.5, 133.7, 131.4 (d, *J* = 7.2 Hz), 129.8 (q, *J* = 34.6 Hz), 129.1, 128.8, 126.8, 126.6, 126.1, 124.7, 122.5 (q, *J* = 272.8 Hz), 119.6 (d, *J* = 7.3 Hz), 116.3 (d, *J* = 21.8 Hz), 50.7, 37.2; HRMS-ESI (*m*/*z*) calcd [M+H]^+^ for C_20_H_13_F_4_N_4_O_3_S_2_^+^ 497.0360, found 497.0358.

*2-{[(2-(4-fluorophenyl)thiazol-4-yl)methyl](methyl)amino}-BTZ* (**36**). Solid, white color; ^1^H NMR indicates 2:3 atropisomeric ratio through the integral value of −CH_2_ protons and the thiazole −CH protons; ^1^H NMR (400 MHz, DMSO-*d*_6_) *δ:* 8.86 (s, 1H), 8.82 (s, 1H), 7.92 (br, 2H), 7.80 (s, 0.4 H, minor), 7.63 (s, 0.6 H, major), 7.49 (br, 2H), 5.18 (s, 1.2H, major), 5.14 (s, 0.8H, minor), 3.47–3.41 (m, 3H); ^13^C NMR (151 MHz, CDCl_3_) *δ:* 166.3, 163.5, 162.3 (d, *J* = 279.0 Hz), 151.1, 144.0, 134.5, 133.7, 130.4, 129.8 (q, *J* = 35.4 Hz), 129.1, 126.8, 126.7, 126.1, 122.5 (d, *J* = 273.1 Hz), 118.2, 116.8, 50.7, 37.2; HRMS-ESI (*m*/*z*) calcd [M+H]^+^ for C_20_H_13_F_4_N_4_O_3_S_2_^+^ 497.0360, found 497.0358.

*2-{methyl[(5-phenyl-1,3,4-oxadiazol-2-yl)methyl]amino}-BTZ* (**37**). Solid, yellow color; ^1^H NMR (400 MHz, CDCl_3_) *δ:* 9.15 (s, 1H), 8.82 (s, 1H), 8.03 (d, *J* = 7.60 Hz, 2H), 7.56–7.46 (m, 3H), 5.43 (s, 2H), 3.54 (s, 3H); ^13^C NMR (151 MHz, CDCl_3_) *δ:* 166.2, 166.1, 164.8, 160.8, 144.0, 133.9 (d, *J* = 2.8 Hz), 132.2, 130.2 (q, *J* = 35.49 Hz), 129.2, 127.1, 126.5, 126.2 (d, *J* = 3.2 Hz), 123.2, 122.3 (q, *J* = 274.0 Hz), 45.2, 36.7; HRMS-ESI (*m*/*z*) calcd [M+H]^+^ for C_19_H_13_F_3_N_5_O_4_S^+^ 464.0635, found 464.0632.

*2-{[(5-(3-fluorophenyl)-1,3,4-oxadiazol-2-yl](methyl)(methyl)amino}-BTZ* (**38**). Solid, yellow color; ^1^H NMR (400 MHz, CDCl_3_) *δ:* 9.14 (s, 1H), 8.81 (s, 1H), 8.03 (t, *J* = 7.0 Hz, 1H), 7.55 (dd, *J* = 6.8 Hz, *J* = 12.4 Hz, 1H), 7.31–7.21 (m, 2H), 5.44 (s, 2H), 3.56 (s, 3H); ^13^C NMR (151 MHz, CDCl_3_) *δ:* 166.1, 165.6, 164.9, 162.8, 160.2 (d, *J* = 259.0 Hz), 144.1, 134.2 (d, *J* = 5.4 Hz), 133.8, 130.3 (q, *J* = 35.8 Hz), 130.0, 126.6, 126.4 (d, *J* = 3.2 Hz), 124.9 (d, *J* = 3.1 Hz), 122.4 (q, *J* = 273.6 Hz), 117.2 (d, *J* = 20.8 Hz), 111.9 (d, *J* = 14.2 Hz), 110.2, 45.3, 37.0; HRMS-ESI (*m*/*z*) calcd [M+H]^+^ for C_19_H_12_F_4_N_5_O_4_S^+^ 482.0541, found 482.0541.

*2-{[(5-(4-fluorophenyl)-1,3,4-oxadiazol-2-yl](methyl)(methyl)amino}-BTZ* (**39**). Solid, yellow color; ^1^H NMR (400 MHz, CDCl_3_) *δ:* 9.14 (s, 1H), 8.82 (s, 1H), 8.04–8.01 (m, 2H), 7.18 (t, *J* = 8.0 Hz, 2H), 5.41 (s, 2H), 3.55 (s, 3H); ^13^C NMR (151 MHz, CDCl_3_) *δ:* 166.1, 165.3, 165.2 (d, *J* = 254.1 Hz), 165.0, 161.0, 144.2, 134.0 (d, *J* = 2.9 Hz), 130.3 (q, *J* = 35.5 Hz), 129.6 (d, *J* = 8.9 Hz), 126.6, 126.4 (d, *J* = 3.2 Hz), 122.4 (q, *J* = 273.6 Hz), 119.7, 116.7 (d, *J* = 22.3 Hz), 45.2, 37.0; HRMS-ESI (*m*/*z*) calcd [M+H]^+^ for C_19_H_12_F_4_N_5_O_4_S^+^ 482.0541, found 482.0541.

*2-{methyl[(5-phenyl-1,3,4-thiadiazol-2-yl)methyl]amino}-BTZ* (**40**). Solid, white color; ^1^H NMR (400 MHz, CDCl_3_) *δ:* 9.17 (s, 1H), 8.82 (s, 1H), 7.93 (d, *J* = 6.8 Hz, 2H), 7.49–7.45 (m, 3H), 5.44 (s, 2H), 3.54 (s, 3H); ^13^C NMR (151 MHz, CDCl_3_) *δ:* 171.2, 165.8, 164.0, 162.1, 134.0, 133.9 (d, *J* = 3.2 Hz), 131.6, 130.3 (q, *J* = 35.6 Hz), 129.8, 129.4, 128.1, 126.6, 126.4 (d, *J* = 3.5 Hz), 122.4 (q, *J* = 273.2 Hz), 110.0, 49.8, 36.9; HRMS-ESI (*m*/*z*) calcd [M+H]^+^ for C_19_H_13_F_3_N_5_O_3_S_2_^+^ 480.0406, found 480.0405.

*2-{[(5-(3-fluorophenyl)-1,3,4-thiadiazol-2-yl](methyl)(methyl)amino}-BTZ* (**41**). Solid, yellow color; ^1^H NMR (400 MHz, CDCl_3_) *δ:* 9.17 (s, 1H), 8.81 (s, 1H), 8.35 (t, *J* = 6.8 Hz, 1H), 7.50 (dd, *J* = 5.6 Hz, *J* = 11.2 Hz, 1H), 7.30 (t, *J* = 7.19 Hz, 1H), 7.20 (t, *J* = 9.6 Hz, 1H), 5.50 (s, 2H), 3.54 (s, 3H); ^13^C NMR (151 MHz, CDCl_3_) *δ:* 170.0, 165.8, 164.1, 163.7, 159.7 (d, *J* = 253.0 Hz), 144.1, 133.9 (d, *J* = 3.0 Hz), 133.1 (d, *J* = 8.4 Hz), 130.3 (q, *J* = 35.6 Hz), 129.2, 126.7, 126.3 (d, *J* = 3.0 Hz), 125.1, 122.3 (q, *J* = 273.1 Hz), 118.1 (d, *J* = 7.7 Hz), 116.6 (d, *J* = 21.7 Hz), 49.5, 36.8; HRMS-ESI (*m*/*z*) calcd [M+H]^+^ for C_19_H_12_F_4_N_5_O_3_S_2_^+^ 498.0312, found 498.0314.

*2-{[(5-(4-fluorophenyl)-1,3,4-thiadiazol-2-yl](methyl)(methyl)amino}-BTZ* (**42**). Solid, yellow color; ^1^H NMR (400 MHz, CDCl_3_) *δ:* 9.17 (s, 1H), 8.82 (s, 1H), 8.02–7.86 (m, 2H), 7.14 (t, *J* = 8.0 Hz, 2H), 5.42 (s, 2H), 3.53 (s, 3H); ^13^C NMR (151 MHz, CDCl_3_) *δ:* 170.1, 165.7, 164.6 (d, *J* = 253.1 Hz), 163.9, 162.1, 144.1, 134.0 (d, *J* = 3.0 Hz), 130.3, 130.1 (d, *J* = 8.7 Hz), 123.0 (q, *J* = 35.4 Hz), 126.5, 126.3 (d, *J* = 3.2 Hz), 126.2, 122.3 (q, *J* = 273.2 Hz), 116.5 (d, *J* = 22.1 Hz), 49.9, 37.0; HRMS-ESI (*m*/*z*) calcd [M+H]^+^ for C_19_H_12_F_4_N_5_O_3_S_2_^+^ 498.0312, found 498.0317.

### 3.2. MIC Determination and Structure Activity Relationship

The compound anti-mycobacterial activity was performed as previously described [11]. The triazole moiety has been widely used in drug molecules and regarded as a privileged motif [12]. More importantly, the facile click chemistry to construct triazole is readily achieved through the reaction between a terminal alkyne and an azide; the 1,2,3-triazole linker compounds **1**–**11** displayed MIC values of 0.009–0.326 µM. The MIC values are shown in Table 1. Specifically, the N-cyclohexyltriazole **1** exhibited an MIC of 0.032 µM against Mtb H37Rv. The N-phenyltriazole **2** was 4-fold more potent, with an MIC of 0.009 µM. Reduced activities were recorded for N-benzyltriazole **3** and compound **4**. Notably, all compounds showed ~5–fold activity improvement as compared to our previously reported 6-methanesulfonyl counterparts [8].

Having profiled the tail position, we then defined the phenyl group and intended to further investigate the substitute effect at the phenyl ring (Table 2), aiming to obtain compounds with higher potency and improved physicochemical properties than compound **2**. Replacement of one or more hydrogens with a fluorine atom is a commonly employed tactic to elevate the compounds’ ADME properties [13]. Thus, the analogs **12**, **13** and **14** with one fluorine atom at different position were prepared. The *meta*-F that substituted compound **13**, with an MIC of 0.008 µM, had the highest potency among these three compounds; compounds **12** and **14** had an MIC of 0.012 µM and 0.270 µM, respectively.

We then explored the meta- position by introducing an electron withdrawing or donating group. Within this frame, compounds **15**–**21** were generated. Unfortunately, none of them exhibited activity improvement; higher MIC values of 0.014–0.207 µM were recorded. Specifically, gradual activity loss was observed from chloro- to bromo- and iodo-. Other groups like methyl, methoxyl, and trifluoromethyl substitution also diminished the Mtb growth inhibition activity. The meta-cyano substituted compound **21** showed the weakest activity. Thus, our optimization choice was turned to phenyl di-substitution, generating compounds **22**–**29**. The di-substitution strategy also failed to provide compounds with further potency increase. Among these compounds, three difluoro-substituted compounds **22**, **23** and **24** showed MICs lower than 0.023 µM, but the 2,5-difluoro substituted compound **24** displayed an MIC of 0.259 µM.

Having established that the triazole linker and the phenyl attachment could drop down MICs to less than 0.01 µM, we next replaced this linker with its bioisosters. Other five-member heterocycles like oxazole, thiazole, oxadiazole or thiodiazole are considered as suitable surrogates of triazole [14]. We postulated that replacement of triazole with the heterocycles likely generates analogs with different physicochemical properties. Compounds **31**–**42** were designed and synthesized using various linkers (Table 3). It was interesting that the oxazole linker compounds **31**–**33** and the three 1,3,4-oxadiazole compounds **37**–**39** displayed antitubercular activity higher than or comparable to compound **2**. The MIC of compound **34** with a thiazole linker was 0.015 µM, while compounds **35** and **36** with substituted phenyl attachment exhibited decreased activity (MIC 0.032 µM and 0.034 µM, respectively). Similarly, the 1,3,4-thiodiazole-linked compounds **40**–**42** displayed higher MIC values.

### 3.3. DprE1 Inhibition Activity

The above compounds with various linkers were selected based on their MIC values to test potential on-target inhibitory activity of the Mtb DprE1 enzyme. Following a previously reported assay method [10], all of the compounds inhibited the DprE1 activity, with the IC_50_ values ranging from 0.02 to 7.2 μM (Figure 2 and Appendix A). Compounds with low MICs accordingly exhibited lower IC_50_ values as observed in the DprE1 inhibitory assays. For example, compounds **2**, **31**, **34**, and **37** had IC_50_ values of 0.02–0.75 μM, and the least potent compound **24** displayed an IC_50_ of 7.2 μM. As a comparison, the positive control PBTZ169 displayed an IC_50_ of 0.009 μM, consistent with the value reported in the literature [6].

### 3.4. Compound-Mediated Cell Wall Inhibition of Mtb

The ability of the compounds **2** and **37** to inhibit the DprE1 enzyme in Mtb H37Rv was examined via [^14^C]-acetate metabolic labeling, as described before (Figure 3) [8,14]. Analysis of the extractable lipids from radiolabeled bacteria by TLC points out to similar profiles for PBTZ169 and the two tested inhibitors, suggesting the same mechanism of action. The cultures treated with the inhibitors accumulated trehalose monomycolates (TMM) and trehalose dimycolates (TDM) compared to the control bacteria, which indicates interference with the build-up of the mycobacterial cell wall core. DprE1 inhibition causes deficiency in the synthesis of arabinan chains serving as attachment sites for mycolic acids, which then incorporate to soluble lipids, TMM and TDM.

### 3.5. Metabolic Stability, Cytotoxicity and Solubility of Selected Compounds

Since compounds **2**, **31**, **34**, **37** and **40** showed higher inhibition potentials against DrpE1, their metabolic stability in human microsomes was then examined (Table 4). As expected, the metabolic stability was affected by the linker portion. Among them, compound **2** containing an 1,2,3-triazole linker had the highest stability (T_1/2_ = 100.4 min) and the lowest intrinsic clearance Cl_int_ (17.31 mL/min/kg), followed by compound **36** with an 1,2,4-oxadiazole linker (T_1/2_ = 68.2 min, and Cl_int_ = 25.48 mL/min/kg). The thiazole linker compound **34** was the least stable.

Compounds **2** and **37** were further examined for cytotoxicity against HepG2 cell using an MTT assay [15]. No inhibition of cell viability was observed at a concentration lower than 1.0 μg/mL (Appendix A). At 5.0 μg/mL, cell viability was reduced to 72–75%. At 16 µg/mL, the compounds inhibitory potency against *E. coli* have been tested and no obvious activity was observed for both compounds.

The compound solubility Is closely related to its bioavailability and in vivo activity, thus, the kinetic solubility of compounds **2** and **37** was determined. In PBS buffer, both compounds exhibited higher solubility than PBTZ169 (Appendix A), which also correlated with their reduced clogP values.

### 3.6. Pharmacokinetics and Efficacy of Compounds **2** and **37**

To further characterize compounds **2** and **37**, we evaluated their pharmacokinetic profiles in mice. BALB/C mice were given compounds **2** or **37** intravenously (i.v., 2 mg/kg) or orally (p.o., 10 mg/kg). As shown in Table 5, both compounds had acceptable PK profiles with good oral bioavailability of 35.7% and 70.6%, respectively. The T_1/2_ and C_max_ values were shorter than PBTZ169 [16], which was dosed orally at 25 mg/kg.

The anti-TB potential of the compounds was evaluated using a mouse model acutely infected with *Mtb* H37Rv. Both compounds were administered: oral dose 50 mg/kg for 4 weeks, once per day, except weekends. PBTZ169 killed Mtb by 3 logs in the lungs compared to the control group (Figure 4). By comparison, compound **37** reduced the bacteria load by 1.2 logs; compound **2** barely displayed efficacy.

## 4. Discussion

BTZ derivatives have been reported to exhibit potent anti-mycobacterial activity in vitro and in vivo and the two analogs, BTZ043 and PBTZ169, are being evaluated in clinical trials [17,18]. However, the poor solubility and high plasma binding percentage for both the compounds indicate that their physicochemical properties are far from optimal [7], thus, continuous efforts for the improvement of the BTZ compounds’ drug-like properties have been undertaken in the research community [19,20].

We previously reported a series of BTZ derivatives with 6-methansulfonyl replacement of 6-CF_3_, along with side chain modification. The optimized BTZ analogs displayed more than 20-fold increased aqueous solubility than PBTZ169 [8]. With MIC values in 10–40 nM ranges and acceptable pharmacokinetic properties, we recently performed the compound efficacy study in the TB-infected mice model. Unfortunately, no obvious CFU reduction (data not reported) was recorded. Nonetheless, our side chain modification strategy highlights alternatives to focus on the linker and tail position; a concerted modification might provide candidates with improved physiochemical properties and high in vivo efficacy.

Herein, we investigated analogs with the original 6-CF_3_ BTZ pharmacophore in combination with our modified side chain moieties. These novel series of compounds exhibited MICs values 5–7-folds lower than that of the corresponding 6-methansulfonyl counterparts, indicating that the original 6-CF_3_ BTZ core structure is more favorable. By performing the DprE1 enzyme activity assay and the cell wall synthesis disruption experiments, we demonstrated the compounds’ on-target inhibitory activity, in accordance with that of PBTZ169. The optimized compound in this novel series of BTZ analogs displayed good PK profile in the mice TB model.

This study demonstrated that our side chain modification approach provided a BTZ derivative with efficacy of reducing the bacteria load in lungs 1.2 logs. Factors influencing the compound in vivo efficacy are complex and we noticed that the PK parameter t_1/2_ for PBTZ 169 was longer than compound **37** [8], that may partially explain the observed potency difference. Our study also showed that the compounds’ metabolic stability was closely related to the linker moiety: the t_1/2_ for the triazole linker compound was 5-folds longer than the thiozole linker, suggesting that further linker exploration might provide more metabolically stable compounds. Finally, the presence of a basic nitrogen in the side chain of PBTZ169 greatly increased the compound solubility in acidic conditions, which facilitated the compound’s gastric absorption [21]. While the lack of such a basic center in neutral compound **37** might be unbeneficial, further investigation by introducing a basic or polar atom at appropriate position might be worthy of being explored.

## 5. Conclusions

In the present work, a new series of BTZ derivatives were prepared. We linked the BTZ core structure with aryl groups using different heterocycles as linkers, including 1,3,4-triazole, oxazole, thiazole, 1,2,4-oxadiazole and 1,2,4-thiadiazole. Compared to the previously reported compounds, these novel BTZ derivatives displayed increased mycobacteria inhibitory activity and metabolic stability. The represented compounds exhibited single digit nanomolar MICs values, and low cytotoxicity against mammalian cells. One compound displayed in vivo efficacy, although it was much worse than the PBTZ169 compound. This study highlighted our side chain modification strategy and expanded the diversity of BTZ compounds. Further linker and side chain modification to increase the compound in vivo potency is going on and will be reported in due course.

## Data Availability

Not applicable.

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
