# Peer review of "Side Chain-Modified Benzothiazinone Derivatives with Anti-Mycobacterial Activity"

_biomedicines, 2023, doi:10.3390/biomedicines11071975_

Round 1

Reviewer 1 Report

This is a decent study with minor limitations were noted. 

I found the study lacks an in vitro cytotoxicity and the observation can enhance the potential of this study. 

Also, please test the compounds with other major pathogens such as ESKAPE pathogens is up to author's discretion. 

The main question addressed by the research is the novel synthesized hybrid molecules and it's activity against MTB.

The topic is original and relevant in the field and it address a specific gap in the field.

The novel molecules and it's potential activity against MTB adds to the subject area compared with other published material.   Cytotoxicity needs to included.    The conclusions are consistent with the evidence and arguments presented and they address the main question posed. The references are appropriate.

Author Response

I found the study lacks an in vitro cytotoxicity and the observation can enhance the potential of this study. 

Response: two representitive compounds have been evaluated for in vitro cytotocicity (manuscript line 603-606). No inhibition of cell viability was observed at concentration lower than 1.0 μg/mL (supporting information, Figure S1). At 5.0 μg/mL, cell viability was reduced to 72-75%.

Also, please test the compounds with other major pathogens such as ESKAPE pathogens is up to author's discretion.

Response:  We previously tested some BTZ derivatives against E.Coli and Staphylococcus aureus, found not inhibitory activities, the result is the same as other published articles(for example: Science, 1999, www.sciencemag.org/cgi/content/full/1171583/DC1). We will test some compounds  to check whether side chain variation would broaden the antibacterial spectrum.

The main question addressed by the research is the novel synthesized hybrid molecules and it's activity against MTB.

The topic is original and relevant in the field and it address a specific gap in the field.

The novel molecules and it's potential activity against MTB adds to the subject area compared with other published material.   Cytotoxicity needs to included.    The conclusions are consistent with the evidence and arguments presented and they address the main question posed. The references are appropriate.

Reviewer 2 Report

The presented article brings interesting results of a series of new substances with antituberculotic activity, which are based on previously published highly effective substances. Unfortunately, the article suffers from some deficiencies that prevent its publication in its current form. I believe that after revision the article will be suitable for publication.

Major comments:

Nomenclature: names of substances do not follow IUPAC rules. Undifferentiated levels of parentheses do not allow for proper reading. Sometimes trivial names (tolyl) are used.

Where atropoisomers occur, spectra for both isomers should be noted separately. The correct names and structures should be given for each form separately.

Chemical synthesis should be in part 2. Materials and methods. On the other hand, yields and spectral characterization of compounds should be in the part 3. Results.

For discussion about structure activity relationships, it is not suitable to use nanograms per ml. Because of different molecular weights of compounds, nanomols per ml should be used, as molecules bind to their targets in molecular ratio.

Minor comments:

Line 56: For Figure 1, general structure of synthesized compounds should be depicted for clarification.

Line 78: List of abbreviations is not complete. HCl is not abbreviation. Because all abbreviations are explained on place where occurs, I recommend to ommit this part.

In scheme 2+3yields should not be in reaction scheme beneath the arrow. It is confusing

Line 677: NMR spectra are presented in Supplementary materials, but are not listed here.

Author Response

The presented article brings interesting results of a series of new substances with antituberculotic activity, which are based on previously published highly effective substances. Unfortunately, the article suffers from some deficiencies that prevent its publication in its current form. I believe that after revision the article will be suitable for publication.

Major comments:

Nomenclature: names of substances do not follow IUPAC rules. Undifferentiated levels of parentheses do not allow for proper reading. Sometimes trivial names (tolyl) are used.

Response: we have added in the abbreviation part to indicate BTZ as the 8-nitro-6-(trifluoromethyl)-4H-benzo[e][1,3]thiazin-4-one pharmacophore. Tolyl is a commonly used name.

Where atropoisomers occur, spectra for both isomers should be noted separately. The correct names and structures should be given for each form separately.

Response: the atropoisomers could not be separated, all compounds displayed a single peak when doing purity analysis using HPLC. But we can observe two different chemical shift in the HNMR spectrum, which we figured comes from the limited bond rotation at 2-position.

Chemical synthesis should be in part 2. Materials and methods. On the other hand, yields and spectral characterization of compounds should be in the part 3. Results.

Response: Chemical synthsis is moved to part 2. Compounds characterization is moved to part 3.

For discussion about structure activity relationships, it is not suitable to use nanograms per ml. Because of different molecular weights of compounds, nanomols per ml should be used, as molecules bind to their targets in molecular ratio.

Response: the unit is changed from ng/mL to uM. Thanks.

Minor comments:

Line 56: For Figure 1, general structure of synthesized compounds should be depicted for clarification.

Response: Compounds in this work are shown in Figure 1.

Line 78: List of abbreviations is no t complete. HCl is not abbreviation. Because all abbreviations are explained on place where occurs, I recommend to ommit this part.

Response: OK, the abbreviation paragraph is deleted.

In scheme 2+3yields should not be in reaction scheme beneath the arrow. It is confusing

Response:  the yield is removed.

Line 677: NMR spectra are presented in Supplementary materials, but are not listed here.

Response: The original NMR spectra are presented in the Supplementary materials. The data are provided in the main manuscript.

Reviewer 3 Report

This paper describes a series of benzothiazinone analogues of PBTZ169 bearing heterocyclic linkers which have improved physiochemical properties. The experimental methods and results are reported clearly, the discussion of the results is presented clearly and logically. It is nice to read a paper which has a good number of novel analogues (enabling a decent discussion of the SAR), MIC and pharmacokinetic data, as well as in vivo studies of selected compounds. 

The compounds are well characterised, thank you for including a comment about rotomers in the 1H NMR spectra of certain compounds in the discussion.

Table 2 describes the antitubercular activity of the novel compounds, and it includes an entry for PBTZ169. Seeing that this paper describes the improved pharmacokinetic parameters of these analogues in comparison to PBTZ169, it might be useful to include the pharmacokinetic parameters for PBTZ169 in tables 4 and 5 for a direct comparison.

In line 540 the sentence should read "the three difluoro-substituted compounds 22, 23 and 25". Compound 25 has an MIC of 7 ng/mL, whereas compound 24 has an MIC of 129 ng/mL, as described in the next sentence.

This paper is well written, but it will require minor grammatical changes before publication, certain sections of this manuscript will require more changes than other parts of the manuscript.

Line 660: thiazole rather than thiozole

Line 664: unbeneficical, needs to be changed to unbeneficial or detrimental.

Author Response

Comments and Suggestions for Authors

This paper describes a series of benzothiazinone analogues of PBTZ169 bearing heterocyclic linkers which have improved physiochemical properties. The experimental methods and results are reported clearly, the discussion of the results is presented clearly and logically. It is nice to read a paper which has a good number of novel analogues (enabling a decent discussion of the SAR), MIC and pharmacokinetic data, as well as in vivo studies of selected compounds. 

The compounds are well characterised, thank you for including a comment about rotomers in the 1H NMR spectra of certain compounds in the discussion.

Table 2 describes the antitubercular activity of the novel compounds, and it includes an entry for PBTZ169. Seeing that this paper describes the improved pharmacokinetic parameters of these analogues in comparison to PBTZ169, it might be useful to include the pharmacokinetic parameters for PBTZ169 in tables 4 and 5 for a direct comparison.

Response: We reported the liver microsome stability of PBTZ169 in our previous paper (J. Med. Chem. 2021, 64, 14526−14539). The pharmacokinetic property of PBTZ169 is much better than compound 2 and 37. That also explains its high in vivo efficacy.

In line 540 the sentence should read "the three difluoro-substituted compounds 22, 23 and 25". Compound 25 has an MIC of 7 ng/mL, whereas compound 24 has an MIC of 129 ng/mL, as described in the next sentence.

Response: this sentence is changed to: Among these compounds, three difluoro-substituted compounds 22, 23 and 24 showed MICs lower than 0.023 µM, but the 2,5-difluoro substituted compound 24 displayed MIC 0.259 µM.

Comments on the Quality of English Language

This paper is well written, but it will require minor grammatical changes before publication, certain sections of this manuscript will require more changes than other parts of the manuscript.

Line 660: thiazole rather than thiozole

Line 664: unbeneficical, needs to be changed to unbeneficial or detrimental.

Response: The two typos are corrected.

Round 2

Reviewer 1 Report

The observation of activity against E. coli can be incorporated in the text. Also, please include the cytotoxicity against Red Blood Cells. 

Author Response

The observation of activity against E. coli can be incorporated in the text. Also, please include the cytotoxicity against Red Blood Cells.

Response: the compounds inhibitory potency against E.coli have been tested and no obvious activity was observed for both compounds. The result is added in the text, line 874.

We consider it necessary to test the compounds activity against both mammalian cell and other bacteria, the reason is to check the compounds selectivity. However, there is no reason to test the compounds cytotoxicity against Red Blood Cells. Especially at this stage when the compounds activity against mycobacteria is not potent enough to be further studied as potential drug candidates.

Reviewer 2 Report

Thanks to the authors for re-shaping the artice. It is much better now, but some serious flaws remains. These can be easily fixed. After that, manuscript will be ready for publication.

Chemical nomenclature is still not correct. Please see IUPAC Blue book chapter P-16.5 Enclosing marks, namely use of parentheses, brackets and braces.

In manuscript for compound 1:

2-(((1-cyclohexyl-1H-1,2,3-triazol-4-yl)methyl)(methyl)amino)-8-nitro-6-(trifluoromethyl)-4H-benzo[e][1,3]thiazin-4-one

Proper name for compound 1 shlould be:

2-{[(1-cyclohexyl-1H-1,2,3-triazol-4-yl)methyl](methyl)amino}-8-nitro-6-(trifluoromethyl)-4H-1,3-benzothiazin-4-one

This problem occures in all names - multiplicated parentheses instead of brackets and braces.

General formula of novel compounds in Figure 1 is not correct - methylene group between tertiary amine and five-membered heterocycle is missing. Marking of heterocycle does not correspond for compounds 1-30 (shifted double-bonds - no double bond from Z carbon, four bonds from Y nitrogen)

Line 823 Supplementary materials please add:

1H and 13C NMR spectra of studied compounds.

Author Response

Thanks to the authors for re-shaping the artice. It is much better now, but some serious flaws remains. These can be easily fixed. After that, manuscript will be ready for publication.

Chemical nomenclature is still not correct. Please see IUPAC Blue book chapter P-16.5 Enclosing marks, namely use of parentheses, brackets and braces.

Response: Thanks. The reviewer suggested parentheses, brackets and braces have been used for the compound names.

General formula of novel compounds in Figure 1 is not correct - methylene group between tertiary amine and five-membered heterocycle is missing. Marking of heterocycle does not correspond for compounds 1-30 (shifted double-bonds - no double bond from Z carbon, four bonds from Y nitrogen)

Response: the methylene group is added in Figure 1. The heterocycles was checked and mistakes are corrected. 

Line 823 Supplementary materials please add:

1H and 13C NMR spectra of studied compounds.

Response: This has been added.

Round 3

Reviewer 2 Report

Authors revised manuscript, now I recommend acceptation.